# SlowFormer: Universal Adversarial Patch for Attack on Compute and Energy Efficiency of Inference Efficient Vision Transformers

## Abstract

Recently, there has been a lot of progress in reducing the computation of deep models at inference time. These methods can reduce both the computational needs and power usage of deep models. Some of these approaches adaptively scale the compute based on the input instance. We show that such models can be vulnerable to a universal adversarial patch attack, where the attacker optimizes for a patch that when pasted on any image, can increase the compute and power consumption of the model. We run experiments with three different efficient vision transformer methods showing that in some cases, the attacker can increase the computation to the maximum possible level by simply pasting a patch that occupies only 8% of the image area. We also show that a standard adversarial training defense method can reduce some of the attack's success. We believe adaptive efficient methods will be necessary for the future to lower the power usage of deep models, so we hope our paper encourages the community to study the robustness of these methods and develop better defense methods for the proposed attack.

## 1 Introduction

The field of deep learning has recently made significant progress in improving the efficiency of inference time. Two broad categories of methods can be distinguished: 1) those that reduce computation regardless of input, and 2) those that reduce the computation depending on the input (adaptively). Most methods, such as weight pruning or model quantization, belong to the first category, which reduces computation by a constant factor, regardless of the input. However, in many applications, the complexity of the perception task may differ depending on the input. For example, when a self-driving car is driving between lanes in an empty street, the perception may be simpler and require less computation when compared to driving in a busy city street scene. Interestingly, in some applications, simple scenes such as highway driving may account for the majority of the time. Therefore, we believe that adaptive computation reduction will become an increasingly important research area in the future, especially when non-adaptive methods reach the lower bound of computation.

We note that reducing computation has at least two advantages: reducing the run time and also reducing the power consumption. We acknowledge that depending on the hardware architecture, reducing the run-time for some input images may not be highly valuable since the system parameters (e.g., camera frame rate) should be designed for the worst-case scenario. Additionally, it might not be possible for other processes to effectively utilize the freed compute cores. However, we argue that reduction of compute usually reduces power usage, which is crucial, particularly in mobile devices that run on battery. This becomes even more important as battery storage technology is not growing as fast as compute technology. For instance, increasing the size of the battery for a drone may lead to a dramatic reduction in its range due to the increased battery weight.

Assuming that a perception method is reducing the computation adaptively with the input, an adversary can trick the model by modifying the input to increase the computation and power consumption. We are interested in designing a universal adversarial patch that when pasted on any input image, will increase the computation of the model leading to increased power consumption. We believe this is an important vulnerability, particularly for safety-critical mobile systems that run on battery.

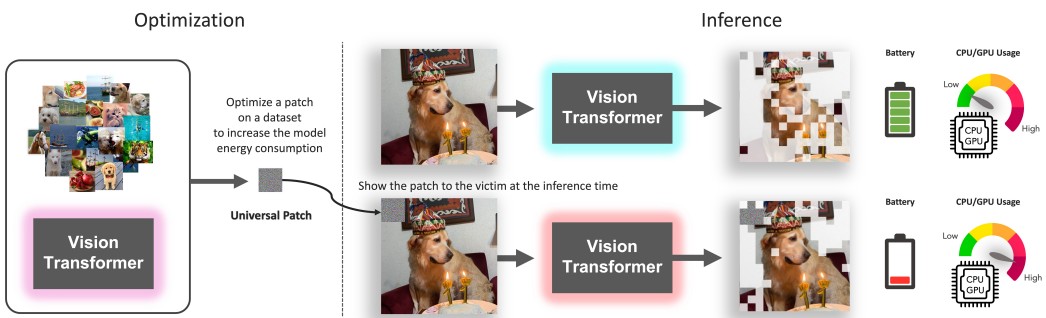

Figure 1: **Energy Attack on Vision Transformers:** Given a pre-trained input-dependent computation efficient model, the adversary first attaches an adversarial patch to all images in a dataset and optimizes this patch with our method such that it maximizes the model's computation for each sample. During inference, the adversary modifies the input of the victim's model by applying the learnt patch to it. This results in an increase in compute in the victim's model. The attack will thus potentially slowdown and also lead to increased energy consumption and CPU/GPU usage on the victim's device.

As an example, a delivery robot like Starship uses a 1,200Wh battery and can run for 12 hours (sta), so it uses almost 100 watts for compute and mobility. Hence, an adversary increasing the power consumption of the perception unit by 20 watts, will reduce the battery life by almost 20%, which can be significant. Note that 20 watts increase in power is realistic assuming that it uses two NVIDIA Jetson Xavier NX cards (almost 20 watts each) to handle its 12 cameras and other sensors.

Please note that in this paper, we do not experiment with real hardware to measure the power consumption. Instead, we report the change in FLOPs of the inference time assuming that the power consumption is proportional to the number of FLOPs.

We design our attack, SlowFormer, for three different methods (A-VIT (Yin et al., 2022), ATS (Fayyaz et al., 2022), and Ada-VIT (Meng et al., 2022)) that reduce the computation of vision transformers. These methods generally identify the importance of each token for the final task and drop the insignificant ones to reduce the computation. We show that in all three cases, our attack can increase the computation by a large margin, returning it to the full-compute level (non-efficient baseline) for all images in some settings. Figure 1 shows our attack.

There are some prior works that design a pixel-level perturbation attack to increase the compute of the model. However, we believe universal patch-based attacks that do not change with the input image (generalize from training data to test data) are much more practical in real applications. Note that to modify the pixel values on a real robot, the attacker needs to access and manipulate the image between the camera and compute modules, which is impossible in many applications.

**Contributions:** We show that efficient vision transformer methods are vulnerable to a universal patch attack that can increase their compute and power usage. We demonstrate this through experiments on three different efficient transformer methods. We show that an adversarial training defense can reduce attack success to some extent.

## 2 RELATED WORK

**Vision Transformers:** The popularity of transformer (Vaswani et al., 2017) architecture in vision has grown rapidly since the introduction of the first vision transformer (Dosovitskiy et al., 2020; Touvron et al., 2021b). Recent works demonstrate the strength of vision transformers on a variety of computer vision tasks (Dosovitskiy et al., 2021; Touvron et al., 2021a; Liu et al., 2021b; Zhou et al., 2021; Rao et al., 2021b; Carion et al., 2020; Zheng et al., 2021; Cheng et al., 2021; Yu et al., 2021; Zhao et al., 2021). Moreover, transformers are the backbone of recent Self-Supervised Learning (SSL) models (He et al., 2021; Caron et al., 2021), and vision-language models (Radford et al., 2021). In our work, we design an attack to target the energy and computation efficiency of vision transformers.

**Efficient Vision Transformers:** Due to the recent importance and popularity of vision transformers, many works have started to study the efficiency of vision transformers (Yu et al., 2022; Brown et al., 2022; Keles et al., 2022). To accomplish this, some lines of work study token pruning with the goal of removing uninformative tokens in each layer (Fayyaz et al., 2022; Rao et al., 2021a; Marin et al., 2021; Yin et al., 2022; Meng et al., 2022). ToMe (Bolya et al., 2022) merges similar tokens in each layer to decrease the computation. Some works address quadratic computation of self-attention module by introducing linear attention (Lu et al., 2021; Katharopoulos et al., 2020; Shen et al., 2021; Ali et al., 2021; Koohpayegani & Pirsiavash, 2022). Efficient architectures (Liu et al., 2021a; Ho et al., 2019) that limit the attention span of each token have been proposed to improve efficiency. In our paper, we attack token pruning based efficient transformers where the computation varies based on the input samples (Meng et al., 2022; Fayyaz et al., 2022; Yin et al., 2022).

**Dynamic Computation:** There are different approaches to reducing the computation of vision models, including knowledge distillation to lighter network (Hinton et al., 2015; Lu et al., 2020), model quantization (Rastegari et al., 2016; Liu et al., 2022) and model pruning (Li et al., 2016). In these methods, the computation is fixed during inference. In contrast to the above models, some works address efficiency by having variable computation based on the input. The intuition behind this direction is that not all samples require the same amount of computation. Several recent works have developed models that dynamically exit early or skip layers (Huang et al., 2017; Teerapittayanon et al., 2016; Bolukbasi et al., 2017; Graves, 2016; Wang et al., 2018; Veit & Belongie, 2018; Guan et al., 2017; Elbayad et al., 2019; Figurnov et al., 2017) and selectively activate neurons, channels or branches for dynamic width (Cai et al., 2021; Fedus et al., 2021; Yuan et al., 2020; Hua et al., 2019; Gao et al., 2018; Herrmann et al., 2020; Bejnordi et al., 2019; Chen et al., 2019) depending on the complexity of the input sample. Zhou et al. show that not all locations in an image contribute equally to the predictions of a CNN model (Zhou et al., 2016), encouraging a new line of work to make CNNs more efficient through spatially dynamic computation. Pixel-Wise dynamic architectures (Ren et al., 2018; Fan et al., 2019; Kong & Fowlkes, 2019; Cao et al., 2019; Verelst & Tuytelaars, 2020; Xie et al., 2020; Chen et al., 2021a) learn to focus on the significant pixels for the required task while Region-Level dynamic architectures perform adaptive inference on the regions or patches of the input (Li et al., 2017; Fu et al., 2017). Finally, lowering the resolution of inputs decreases computation, but at the cost of performance. Conventional CNNs process all regions of an image equally, however, this can be inefficient if some regions are "easier" to process than others (Howard et al., 2017). Correspondingly, (Yang et al., 2020; 2019) develop methods to adaptively scale the resolution of images.

Transformers have recently become extremely popular for vision tasks, resulting in the release of a few input-dynamic transformer architectures (Yin et al., 2022; Fayyaz et al., 2021; Meng et al., 2022). Fayyaz et al. (Fayyaz et al., 2021) introduce a differentiable parameter-free Adaptive Token Sampler (ATS) module which scores and adaptively samples significant tokens. ATS can be plugged into any existing vision transformer architecture. A-VIT (Yin et al., 2022) reduces the number of tokens in vision transformers by discarding redundant spatial tokens. Meng et al. (Meng et al., 2022) propose AdaViT, which trains a decision network to dynamically choose which patch, head, and block to keep/activate throughout the backbone.

**Adversarial Attack:** Adversarial attacks are designed to fool models by applying a targeted perturbation or patch on an image sample during inference (Szegedy et al., 2013; Goodfellow et al., 2014; Kurakin et al., 2018). These methods can be incorporated into the training set and optimized to fool the model. Correspondingly, defenses have been proposed to mitigate the effects of these attacks (Papernot et al., 2016; Xie et al., 2017; Feinman et al., 2017; Li & Li, 2017). Patch-Fool (Fu et al., 2022) considers adversarial patch-based attacks on transformers. Most prior adversarial attacks target model accuracy, ignoring model efficiency.

**Energy Attack:** Very recently, there have been a few works on energy adversarial attacks on neural networks. In ILFO (Haque et al., 2020), Haque et al. attack two CNN-based input-dynamic methods: SkipNet (Wang et al., 2018) and SACT (Figurnov et al., 2017) using image specific perturbation. DeepSloth (Hong et al., 2020) attack focuses on slowing down early-exit methods, reducing their energy efficiency by 90-100%. GradAuto (Pan et al., 2022) successfully attacks methods that are both dynamic width and dynamic depth. NICGSlowDown and TransSlowDown (Chen et al., 2022; 2021b) attack neural image caption generation and neural machine translation methods, respectively. All these methods primarily employ image specific perturbation based adversarial attack. SlothBomb injects efficiency backdoors to input-adaptive dynamic neural networks (Chen et al.) and NodeAttack

(Haque et al., 2021) attacks Neural Ordinary Differential Equation models, which use ordinary differential equation solving to dynamically predict the output of a neural network. Our work is closely related to ILFO (Haque et al., 2020), DeepSloth (Hong et al., 2020) and GradAuto (Pan et al., 2022) in that we attack the computational efficiency of networks. However, unlike these methods, we focus on designing an adversarial patch-based attack that is universal and on vision transformers. We additionally provide a potential defense for our attack. We use a patch that generalizes from train to test set and thus we do not optimize per sample during inference. Our patch-based attack is especially suited for transformer architectures (Fu et al., 2022).

## 3 ENERGY ATTACK

### 3.1 THREAT MODEL:

We consider a scenario where the adversary has access to the victim's trained deep model and modifies its input such that the energy consumption of the model is increased. To make the setting more practical, instead of perturbing the entire image, we assume that the adversary can modify the input image by only pasting a patch (Brown et al., 2017; Saha et al., 2019) on it and that the patch is universal, that is, image independent. During inference, a pretrained patch is pasted on the test image before propagating it through the network.

In this paper, we attack three state-of-the-art efficient transformers. Since the attacker manipulates only the input image and not the network parameters, the attacked model must have dynamic computation that depends on the input image. As stated earlier, several recent works have developed such adaptive efficient models and we believe that they will be more popular in the future due to the limits of non-adaptive efficiency improvement.

### 3.2 ATTACK ON EFFICIENT VISION TRANSFORMERS:

**Universal Adversarial Patch:** We use an adversarial patch to attack the computational efficiency of transforms. The learned patch is universal, that is, a single patch is trained and is used during inference on all test images. The patch optimization is performed only on the train set and there is no per-sample optimization on the test images. The patch is pasted on an image by replacing the image pixels using the patch. We assume the patch location does not change from train to test. The patch pixels are initialized using IID samples from a uniform distribution over $[0, 255]$. During each training iteration, the patch is pasted on the mini-batch samples and is updated to increase the computation of the attacked network. The patch values are projected onto $[0, 255]$ and quantized using 256 uniform levels after each iteration. Note that the parameters of the network being attacked are not updated during patch training. During inference, the trained patch is pasted on the test images and the computational efficiency of the network on the adversarial image is measured. Below, we describe in detail the efficient methods under attack and the loss formulation used to update the patch.

Here, we focus on three methods employing vision transformers for the task of image classification. All these methods modify the computational flow of the network based on the input image for faster inference. A pretrained model is used for the attack and is not modified during our adversarial patch training. For clarity, we first provide a brief background of each method before describing our attack.

**Attacking A-ViT :**

**Background:** A-ViT (Yin et al., 2022) adaptively prunes image tokens to achieve speed-up in inference with minimal loss in accuracy. For a given image, a dropped token will not be used again in the succeeding layers of the network. Let $x$ be the input image and $\{t^l\}_{1:K}$ be the corresponding $K$ tokens at layer $l$. An input-dependent halting score $h_k^l$ for a token $k$ at layer $l$ is calculated and the token is dropped at layer $N_k$ where its cumulative halting score exceeds a fixed threshold value $1 - \epsilon$ for the first time. The token is propagated until the final layer if its score never exceeds the threshold. Instead of introducing a new activation for $h_k^l$, the first dimension of each token is used to predict the halting score for the corresponding token. The network is trained to maximize the cumulative halting score at each layer and thus drop the tokens earlier. The loss termed ponder loss, is given by:

$$\mathcal{L}_{\text{ponder}} = \frac{1}{K} \sum_{k=1}^{K} (N_k + r_k), \qquad r_k = 1 - \sum_{l=1}^{N_{k-1}} h_k^l \qquad (1)$$

Additionally, A-ViT enforces a Gaussian prior on the expected halting scores of all tokens via $KL$-divergence based distribution loss, $\mathcal{L}_{\text{distr.}}$. These loss terms are minimized along with the task-specific loss $\mathcal{L}_{\text{task}}$. Thus, the overall training objective is $\mathcal{L} = \mathcal{L}_{\text{task}} + \alpha_d \mathcal{L}_{\text{distr.}} + \alpha_p \mathcal{L}_{\text{ponder}}$ where $\alpha_d$ and $\alpha_p$ are hyperparameters.

**Attack:** Here, we train the patch to increase the inference compute of a trained A-ViT model. Since we are interested in the compute and not task-specific performance, we simply use $-(\alpha_d \mathcal{L}_{\text{distr.}} + \alpha_p \mathcal{L}_{\text{ponder}})$ as our loss. It is possible to preserve (or hurt) the task performance by additionally using $+\mathcal{L}_{\text{task}}$ (or $-\mathcal{L}_{\text{task}}$) in the loss formulation.

**Attacking AdaViT:**

**Background:** To improve the inference efficiency of vision transformers, AdaViT (Meng et al., 2022) inserts and trains a decision network before each transformer block to dynamically decide which patches, self-attention heads, and transformer blocks to keep/activate throughout the backbone. The $l^{\text{th}}$ block's decision network consists of three linear layers with parameters $W_l = W_l^p, W_l^h, W_l^b$ which are then multiplied by each block's input $Z_l$ to get $m$.

$$(m_l^p, m_l^h, m_l^b) = (W_l^p, W_l^h, W_l^b)Z_l \tag{2}$$

The value $m$ is then passed to sigmoid function to convert it to a probability value used to make the binary decision of keep/discard. Gumbel-Softmax trick (Maddison et al., 2016) is used to make this decision differentiable during training. Let $M$ be the keep/discard mask after applying Gumbel-Softmax on $m$. The loss on computation is given by:

$$\mathcal{L}_{usage} = \left(\frac{1}{D_p}\sum_{d=1}^{D_p} M_d^p - \gamma_p\right)^2 + \left(\frac{1}{D_h}\sum_{d=1}^{D_h} M_d^h - \gamma_h\right)^2 + \left(\frac{1}{D_b}\sum_{d=1}^{D_b} M_d^b - \gamma_b\right)^2 \tag{3}$$

where $D_p, D_h, D_b$ represent the number of total patches, heads, and blocks of the entire transformer, respectively. $\gamma_p, \gamma_h, \gamma_b$ denote the target computation budgets i.e. the percentage of patches/heads/blocks to keep. The total loss is a combination of task loss (cross-entropy) and computation loss: $L = \mathcal{L}_{ce} + \mathcal{L}_{usage}$.

**Attack:** To attack this model, we train the patch to maximize the computation loss $\mathcal{L}_{usage}$. More specifically, we set the computation-target $\gamma$ values to 0 and negate the $\mathcal{L}_{usage}$ term in Eq. 3. As a result, the patch is optimized to maximize the probability of keeping the corresponding patch (p), attention head (h), and transformer block (b). We can also choose to attack the prediction performance by selectively including or excluding the $\mathcal{L}_{ce}$ term. Note that the computation increase for this method is not as high as for the other methods. To investigate, we ran a further experiment using a patch size of 224x224 (entire image size) to find the maximum possible computation. This resulted in 4.18 GFLOPs on the ImageNet-1K validation set, which is lower than 4.6. If we use this as an upper-bound of GFLOPs increase, our method instead achieves a 49% Attack Success.

**Attacking ATS:**

**Background:** Given $N$ tokens with the first one as the classification token, the transformer attention matrix $\mathcal{A}$ is calculated by the following dot product: $\mathcal{A} = \text{Softmax}\left(\mathcal{Q}\mathcal{K}^T/\sqrt{d}\right)$ where $\sqrt{d}$ is a scaling coefficient, $d$ is the dimension of tokens, $\mathcal{Q}, \mathcal{K}$ and $\mathcal{V}$ are the query, key and value matrices, respectively. The value $\mathcal{A}_{1,j}$ denotes the attention of the classification token to token $j$. ATS (Fayyaz et al., 2022) assigns importance score $S_j$ for each token $j$ by measuring how much the classification token attends to it:

$$S_j = \frac{\mathcal{A}_{1,j} \times ||\mathcal{V}_j||}{\sum_{i=2} \mathcal{A}_{1,i} \times ||\mathcal{V}_i||} \tag{4}$$

The importance scores are converted to probabilities and are used to sample tokens, where tokens with a lower score have more of a chance of being dropped.

**Attack:** Since ATS uses inverse transform sampling, it results in fewer samples if the importance distribution is sharp. To maximize the computation in ATS, we aim to obtain a distribution of scores

with high entropy to maximize the number of retained tokens. Therefore, we optimize the patch so that the attention of the classification token over other tokens is a uniform distribution using the following MSE loss:

$$\mathcal{L} = \sum_{i=2}^{N} ||\mathcal{A}_{1,i} - \frac{1}{N}||_2^2$$

Note that one can optimize $\mathcal{S}$ to be uniform, but we found the above loss to be easier to optimize. For a multi-head attention layer, we calculate the loss for each head and then sum the loss over all heads. Moreover, ATS can be applied to any layer of vision transformers. For a given model, we apply our loss at all ATS layers and use a weighted summation for optimization.

## 4 DEFENSE

We adopt standard adversarial training as a defense method for our attack. In the standard way, at each iteration of training the model, one would load an image, attack it, and then use it with correct labels in training the model. We cannot adopt this out-of-the-box since our attack generalizes across images and is not dependent on a single image only. To do this, we maintain a set of adversarial patches, and at each iteration sample one of them randomly (uniformly), and use it at the input while optimizing the original loss of the efficient model to train a robust model. To adapt the set of adversarial patches to the model being trained, we interrupt the training at every 20% mark of each epoch and optimize for a new patch to be added to the set of patches. To limit the computational cost of training, we use only 500 iterations to optimize for a new patch, which results in an attack with reasonable accuracy compared to our main results.

## 5 EXPERIMENTS

### 5.1 ATTACK ON EFFICIENT VISION TRANSFORMERS

**Dataset:** We evaluate the effectiveness of our attack on two datasets: ImageNet-1K (Deng et al., 2009) and CIFAR-10 (Krizhevsky, 2009). ImageNet-1K contains 1.3M images in the train set and 50K images in the validation set with 1000 total categories. CIFAR-10 has 50K images for training and 10K images for validation with 10 total categories.

**Metrics:** We report Top-1 accuracy and average computation in terms of GFLOPs for both attacked and unattacked models. Similar to Attack Success Rate in a standard adversarial attack, we introduce a metric: Attack Success to quantify the efficacy of the attack. We define Attack Success as the number of FLOPs increased by the attack divided by the number of FLOPs decreased by the efficient method. Attack Success $= \frac{(\text{FLOPs}_{\text{attack}} - \text{FLOPs}_{\text{min}})}{(\text{FLOPs}_{\text{max}} - \text{FLOPs}_{\text{min}})}$ where $\text{FLOPs}_{\text{min}}$ is the compute of the efficient model and $\text{FLOPs}_{\text{max}}$ is that of the original inefficient model. Attack Success is thus capped at $100\%$ while a negative value denotes a reduction in FLOPs. Note that our Attack Success metric illustrates the effectiveness of an attack in reversing the FLOPs reduction of a particular method.

**Baselines:** We propose three alternative approaches to SlowFormer (ours) to generate the patch.

**Random Patch:** A simple baseline is to generate a randomly initialized patch. We sample IID pixel values from a uniform distribution between $0$ and $255$ to create the patch.

**NTAP:** We consider a standard adversarial patch that is trained to attack the model task performance instead of compute. We use a non-targeted universal adversarial patch (NTAP) to attack the model. We train the patch to fool the model by misclassifying the image it is pasted on. We use the negative of the cross-entropy loss with the predicted and ground-truth labels as the loss to optimize the patch.

**TAP:** For this baseline, we train a universal targeted adversarial patch (TAP). The patch is optimized to classify all images in the train set to a single fixed category. Similar to NTAP, the adversarial attack here is on task performance and not computation. We experiment with ten randomly generated target category labels and report the averaged metrics.

Table 1: **Energy Attack on Efficient Vision Transformers:** Comparison of the effect of energy attack with baselines: No Attack, Random Patch, targeted (TAP), and non-targeted (NTAP) adversarial patches applied to three input-dynamic computation efficient pre-trained models of varying architectures. The maximum possible compute for a given architecture is provided in bold. On A-ViT , we completely undo the efficiency gains obtained by the efficient method through our attack, achieving Attack Success of 100%. We achieve high Attack Success on all approaches while the baselines expectedly do not contribute to increase in compute.

Figure 2: **Visualization of our Energy Attack on Vision Transformers:** We visualize the A-ViT-Small with and without our attack. We use patch size of 32 for the attack (on the top-left corner). We show pruned tokens at layer 8 of A-ViT-Small. Our attack can recover most of the pruned tokens, resulting in increased computation and power consumption. Note that although the patch is reasonably small and is in the corner of the view, it can affect the whole computational flow of the network. This is probably due to the global attention mechanism in transformers.

| Method | Attack | Model GFLOPs | Top-1 Acc | Attack Success |
|---|---|---|---|---|
| | **ViT-Tiny** | **1.3** | - | - |
| | No attack | 0.87 | 71.4% | - |
| | Random Patch | 0.87 | 70.8% | -1% |
| A-ViT | TAP | 0.85 | 0.1% | -5% |
| | NTAP | 0.83 | 0.1% | -10% |
| | SlowFormer (ours) | 1.3 | 4.7% | 100% |
| | **ViT-Small** | **4.6** | - | - |
| | No attack | 3.7 | 78.8% | - |
| | Random Patch | 3.7 | 78.4% | -2% |
| A-ViT | TAP | 3.6 | 0.1% | -12% |
| | NTAP | 3.6 | 0.1% | -7% |
| | SlowFormer (ours) | 4.6 | 2.3% | 99% |
| | **ViT-Tiny** | **1.3** | - | - |
| | No attack | 0.84 | 70.3% | - |
| | Random Patch | 0.83 | 69.8% | -2% |
| ATS | TAP | 0.76 | 0.1% | -17% |
| | NTAP | 0.61 | 0.1% | -50% |
| | SlowFormer (ours) | 1.0 | 1.2% | 35% |
| | **ViT-Small** | **4.6** | - | - |
| | No attack | 3.1 | 79.2% | - |
| | Random Patch | 3.1 | 78.6% | -1% |
| ATS | TAP | 3.0 | 0.1% | -7% |
| | NTAP | 2.4 | 0.1% | -47% |
| | SlowFormer (ours) | 4.0 | 1.0% | 60% |
| | **ViT-Base** | **17.6** | - | - |
| | No attack | 12.6 | 81.3% | - |
| | Random Patch | 12.5 | 81.2% | -2% |
| ATS | TAP | 12.0 | 0.1% | -12% |
| | NTAP | 11.0 | 0.1% | -32% |
| | SlowFormer (ours) | 15.4 | 0.2% | 52% |
| | **ViT-Small** | **4.6** | - | - |
| | No attack | 2.25 | 77.3% | - |
| | Random Patch | 2.20 | 76.9% | -2% |
| AdaViT | TAP | 2.28 | 0.1% | 1% |
| | NTAP | 2.15 | 0.1% | -4% |
| | SlowFormer (ours) | 3.2 | 0.4% | 40% |

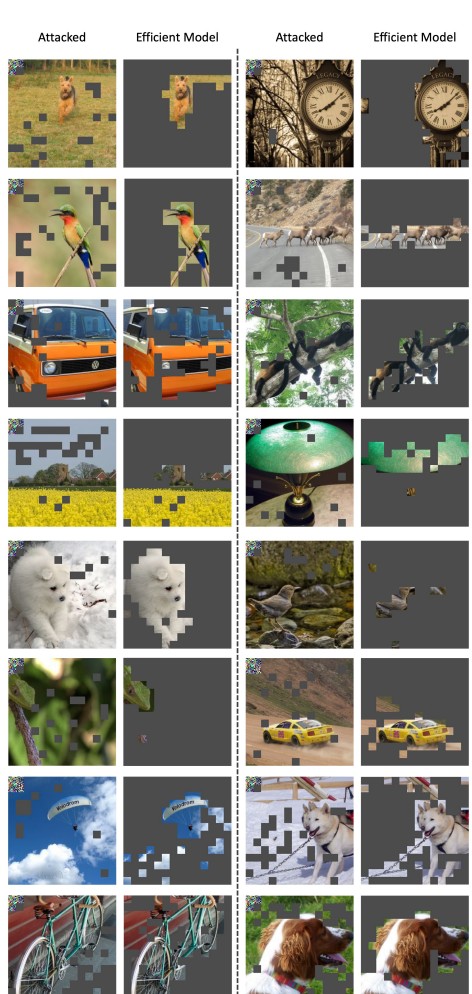

**Implementation Details:** We use PyTorch (Paszke et al., 2019) for all experiments. Unless specified, we use a patch of size $64 \times 64$, train and test on $224 \times 224$ images, and we paste the patch on the top-left corner. Note that our patch occupies just 8% of the total area of an input image. We use AdamW (Loshchilov & Hutter, 2019) optimizer to optimize the patches and use 4 NVIDIA RTX 3090 GPUs for each experiment. We use varying batch sizes and learning rates for each of the computation-efficient methods.

**ATS Details:** For our experiments on ATS, we use the weights of the DeiT model and replace regular attention blocks with the ATS block without training. As in ATS (Fayyaz et al., 2022), we replace layers 3 through 9 with the ATS block and set the maximum limit for the number of tokens sampled

Table 2: **Results on CIFAR10 dataset.** We report results on CIFAR10 dataset to show that our attack is not specific to ImageNet alone. CIFAR-10 is a small dataset compared to ImageNet and thus results in an extremely efficient A-ViT model. Our attack increases the FLOPs from 0.11 to 0.58 which restores nearly 41% of the original reduction in the FLOPs.

| Method | Model FLOPs | Top-1 Acc | Attack Success |
|---|---|---|---|
| ViT-Tiny | 1.26 | - | - |
| A-ViT-Tiny | 0.11 | 95.8% | - |
| SlowFormer (ours) | 0.58 | 60.2% | 41% |

Table 3: **Accuracy controlled compute adversarial attack:** We attack the the efficiency of A-ViT while either maintaining or destroying its classification performance. We observe that our attack can achieve a huge variation in task performance without affecting the Attack Success. The ability to attack the computation without affecting the task performance might be crucial in some applications.

| Attack | Model GFLOPs | Attack Success | Top-1 Acc |
|---|---|---|---|
| ViT-Tiny | 1.26 | - | - |
| No attack | 0.87 | - | 71.4% |
| Acc agnostic | 1.26 | 100% | 4.7% |
| Preserve acc | 1.23 | 92% | 68.5% |
| Destroy acc | 1.26 | 100% | 0.1% |

to 197 for each layer. We train the patch for 2 epochs with a learning rate of 0.4 for ViT-Tiny and $lr = 0.2$ for ViT-Base and ViT-Small.

**A-ViT Details:** We show results on ViT-Tiny and ViT-Small architectures using pretrained models provided by the authors of A-ViT . The patches are optimized for one epoch with a learning rate of 0.2 and batch size of 128. For the training of adversarial defense, we generate 5 patches per epoch of adversarial training and limit the number of iterations for patch generation to 500.

**AdaViT Details:** The authors of AdaViT provide a DeiT-S model pre-trained with their method. For this architecture, we freeze the weights and optimize for our adversarial patch. We use a $lr = 0.2$ and a batch size of 128. Additional details on all three efficient approaches are provided in the supplementary material.

**Results.** The results of our attack, SlowFormer , on various methods on ImageNet dataset are shown in table 1. In A-ViT, we successfully recover 100% of the computation reduced by A-ViT . Our attack has an Attack Success of 60% on ATS and 40% on AdaViT with ViT-Small. A random patch attack has little effect on both the accuracy and computation of the method. Both standard adversarial attack baselines, TAP and NTAP, reduce the accuracy to nearly 0%. Interestingly, these patches further decrease the computation of the efficient model being attacked. This might be because of the increased importance of adversarial patch tokens to the task and thus reduced importance of other tokens. Targeted patch (TAP) has a significant reduction in FLOPs on the ATS method. Since the token dropping in ATS relies on the distribution of attention values of classification tokens, a sharper distribution due to the increased importance of a token can result in a reduction in computation.

We report the results on CIFAR-10 dataset in Table 2. The efficient model (A-ViT ) drastically reduces the computation from 1.26 GFLOPs to 0.11 GFLOPs. Most of the tokens are dropped as early as layer two in the efficient model. SlowFormer is able to effectively attack even in such extreme scenarios, achieving an Attack Success of 40% and increasing the mean depth of tokens from nearly one to five.

We additionally visualize the effectiveness of our attack in Figure 3. The un-attacked efficient method retains only highly relevant tokens at the latter layers of the network. However, our attack results in nearly the entire image being passed through all layers of the model for all inputs. In Fig. 3, we visualize the optimized patches for each of the three efficient methods.

## 5.2 ABLATIONS:

We perform all ablations on the A-ViT approach using their pretrained ViT-Tiny architecture model.

**Accuracy controlled compute adversarial attack:** As we show in Table 1, our attack can not only increase the computation, but also reduce the model accuracy. This can be desirable or hurtful based on the attacker's goals. A low-accuracy model might be an added benefit, similar to regular adversaries, but might also lead to the victim detecting the attack. Here, we show that it is possible to

Table 4: **Effect of patch size:** Analysis of the effect of adversarial patch size on the attack success rate on A-ViT. Our attach is reasonably successful even using a small patch size ($32 \times 32$), which is only 2% of the image area. Interestingly, a small patch on the corner of the view affects the computational flow of the entire transformer model. This might be due to the global attention mechanism in transformers.

| Patch Size (Area) | Model GFLOPs | Top-1 Accuracy | Attack Success |
|---|---|---|---|
| ViT-Tiny | 1.26 | - | - |
| A-ViT-Tiny | 0.87 | 71.4% | - |
| 64 (8%) | 1.26 | 4.7% | 100% |
| 48 (5%) | 1.26 | 1.8% | 99% |
| 32 (2%) | 1.22 | 17.4% | 90% |
| 16 (0.5%) | 0.98 | 63.3% | 27% |
| ViT-Small | 4.6 | - | - |
| A-ViT-Small | 3.7 | 78.8% | - |
| 64 (8%) | 4.6 | 2.3% | 99% |
| 48 (5%) | 4.6 | 5.1% | 98% |
| 32 (2%) | 4.4 | 39.5% | 78% |
| 16 (0.5%) | 3.8 | 78.2% | 16% |

Table 5: **Defense using adversarial training:** We propose and show the impact of our defense for our adversarial attack on A-ViT . Our defense is simply maintaining a set of universal patches and training the model to be robust to a random sample of those at each iteration. The defense reduces the computation to some extent (1.26 to 1.01), but is still far from the original unattacked model (0.87).

| Method | GFLOPs | Top-1 Acc. | Attack Success |
|---|---|---|---|
| No attack | 0.87 | 71.4 | - |
| SlowFormer | 1.26 | 4.7% | 100% |
| Adv Defense + SlowFormer | 1.01 | 65.8% | 34% |

| A-ViT | ATS | AdaViT |
|---|---|---|

Figure 3: **Visualization of optimized patch:** We show the learned universal patches for each of the three efficient methods.

attack the computation of the model while either preserving or destroying the task performance by additionally employing a task loss in the patch optimization. As seen in Table 3, the accuracy can be significantly modified while maintaining a high Attack Success.

**Effect of patch size:** We vary the patch size from $64 \times 64$ to $16 \times 16$ (just a single token) and report the results in Table 4. Interestingly, our attack with ViT-Small has a 73% Attack Success with a $32 \times 32$ patch size, which occupies only 2% of the input image area.

**Effect of patch location:** We vary the location of the patch to study the effect of location on the Attack Success. We randomly sample a location in the image to paste the patch on. We perform five such experiments and observe an Attack Success of 100% for all patch locations.

## 5.3 ADVERSARIAL TRAINING BASED DEFENSE

Our simple defense that is adopted from standard adversarial training is explained in Section 4. The results for defending against attacking A-ViT are shown in Table 5. The original A-ViT reduces the GFLOPs from 1.26 to 0.87, our attack increases it back to 1.26 with 100% attack success. The proposed defense reduces the GFLOPs to 1.01 which is still higher than the original 0.87. We hope our paper encourages the community to develop better defense methods to reduce the vulnerability of efficient vision transformers.

## 6 CONCLUSION

Recently, we have seen efficient transfer models in which the computation is adaptively modified based on the input. We argue that this is an important research direction and that there will be more progress in this direction in the future. However, we show that the current methods are vulnerable to a universal adversarial patch that increases the computation and thus power consumption at inference time. Our experiments show promising results for three SOTA efficient transformer models, where a small patch that is optimized on the training data can increase the computation to the maximum possible level in the testing data in some settings. We also propose a defense that reduces the effectiveness of our attack. We hope our paper will encourage the community to study such attacks and develop better defense methods.

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

## A APPENDIX

In sections C and B, we provide visualizations of our learned patches and token dropping respectively. In Sec. D, we provide additional details on our train and test settings. We also provide the code for our implementation with the default hyperparameters as part of the supplementary material.

## B PATCH VISUALIZATION

We optimize patches for A-ViT using different initializations and visualize them in Fig. 4. All patches achieve Attack Success close to 100%. Presence of multiple universal adversarial patches highlights the vulnerability of the current efficient methods.

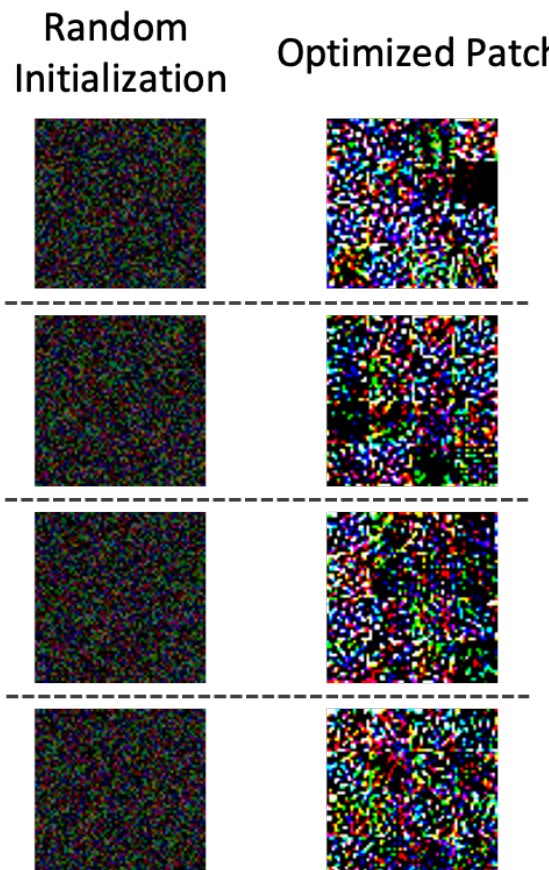

Figure 4: **Optimized patches With different initializations:** Here, we show the optimized patches for A-ViT . A different initialization is used to train each of these patches. All patches achieve Attack Success close to 100%. Presence of multiple universal adversarial patches highlights the vulnerability of the current efficient methods.

We show the evolution of the patch as training progresses in Fig. 5. The patch is trained to attack A-ViT approach. We observe that the patch converges quickly, requiring less than an epoch for 100% Attack Success. The patch at 1000 iterations (0.1 epoch) is similar to that at 10000 iterations (1 epoch) in terms of both appearance and attack performance.

| 10 | 100 | 1000 | 2000 | 5000 | 10000 |

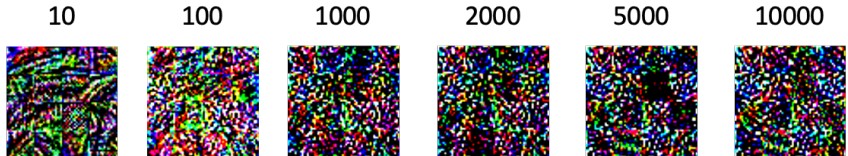

Figure 5: **Visualization of patch optimization:** We train our patch to attack A-ViT and display the patch at various stages of optimization. We observe that the patch converges quickly. The patch at 1000 iterations (0.1 epoch) is similar to that at 10000 iterations (1 epoch) in terms of both appearance and attack performance.

## C  VISUALIZATION OF TOKEN DROPPING

In Fig. 6, we visualize dropped tokens in A-ViT-Small with and without our attack. Our attack significantly decreases the number of pruned tokens, resulting in more compute and energy consumption for the efficient transformer model.

## D  IMPLEMENTATION DETAILS

**ATS Details:**  As in ATS Fayyaz et al. (2022), we replace layers 3 through 9 of ViT networks with the ATS block and set the maximum limit for the number of tokens sampled to 197 for each layer. We train the patch for 2 epochs with a learning rate of $0.4$ for ViT-Tiny and $lr = 0.2$ for ViT-Base and ViT-Small. We use a batch size of $1024$ and different loss coefficients for each layer of ATS. For DeiT-Tiny we use $[1.0, 0.2, 0.2, 0.2, 0.01, 0.01, 0.01]$, for DeiT-Small we use $[1.0, 0.2, 0.05, 0.01, 0.005, 0.005, 0.005]$, and for DeiT-Base we use $[2.0, 0.1, 0.02, 0.01, 0.005, 0.005, 0.005]$ The weights are vastly different at initial and final layers to account for the difference in loss magnitudes across layers.

**A-ViT Details:**  The patches are optimized for one epoch with a learning rate of $0.2$ and a batch size of $512$ ($128 \times 4$GPUs) using AdamW Loshchilov & Hutter (2019) optimizer. We optimize the patches for 4 epochs for patch length 32 and below. For CIFAR-10 experiments, the images are resized from $32 \times 32$ to $256 \times 256$ and a $224 \times 224$ crop is used as the input. For the training of adversarial defense, we generate 5 patches per epoch of adversarial training and limit the number of iterations for patch generation to $500$. The learning rate for patch optimization is increased to $0.8$ for faster convergence.

**AdaViT Details:**  We use a learning rate of $0.2$ and a batch size of $128$ with 4GPUs for patch optimization. We use AdamW Loshchilov & Hutter (2019) optimizer with no decay and train for 2 epochs with a patch size of 64 x 64. We train on the ImageNet-1k train dataset and evaluate it on the test set.

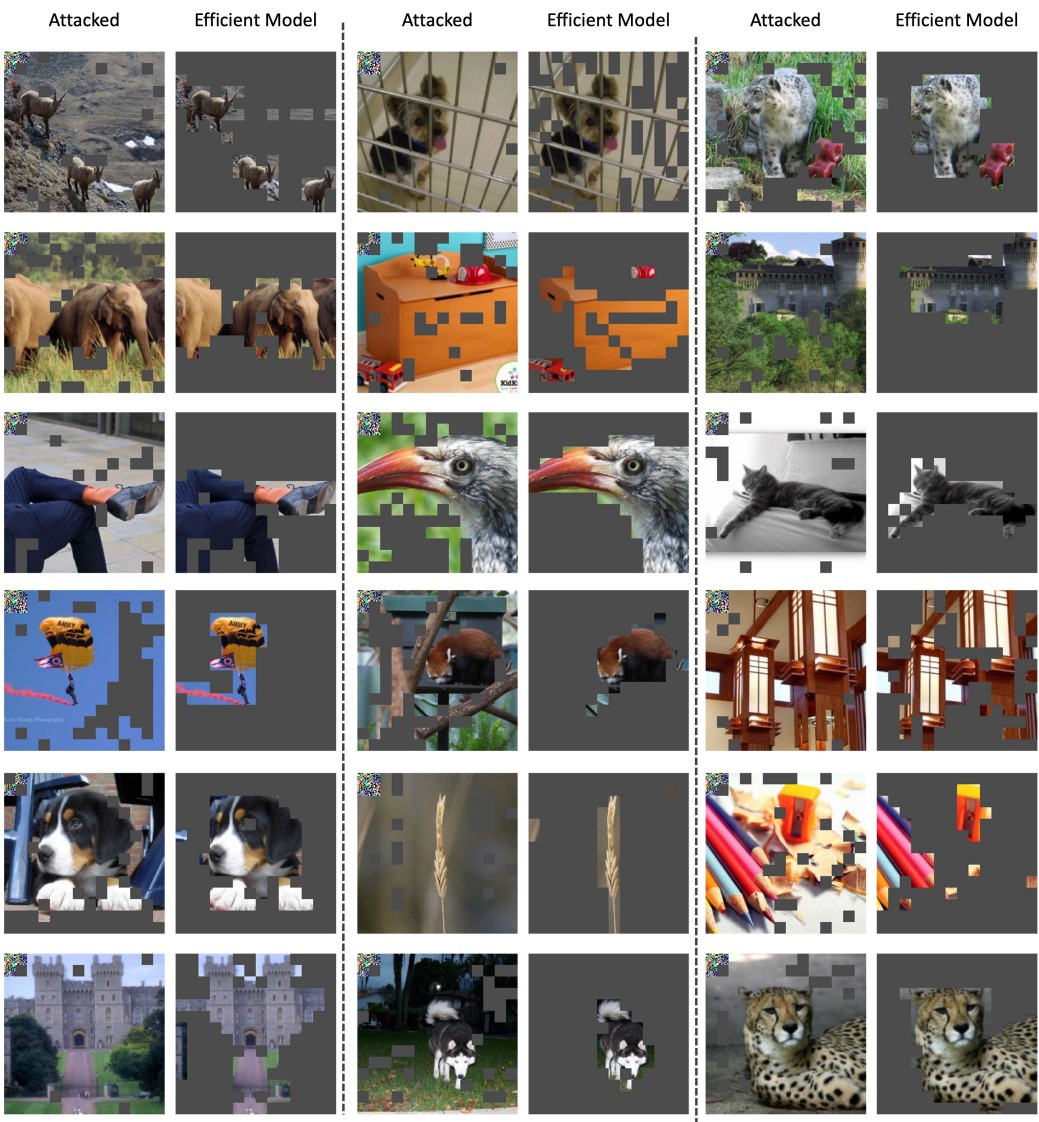

Figure 6: **Visualization of our Energy Attack on Vision Transformers:** Similar to Figure 2 of the main submission, we visualize the A-ViT-Small with and without our attack. We use patch size of 32 for the attack (on the top-left corner). We show pruned tokens at layer 8 of A-ViT-Small. Our attack can recover most of the pruned tokens, resulting in increased computation and power consumption.

