# OpenReview forum: "SlowFormer: Universal Adversarial Patch for Attack on Compute and Energy Efficiency of Inference Efficient Vision Transformers"
_ICLR.cc/2024/Conference — ICLR 2024 Conference Withdrawn Submission_

### Official Review · Reviewer_UkHb · 2023-10-29

**Soundness:** 3 good
**Presentation:** 3 good
**Contribution:** 2 fair
**Rating:** 5
**Confidence:** 5

**Summary:**

This paper proposes SlowFormer which uses adversarial patch attacks to increase the energy consumption of ViT with an adaptive inference procedure. The adversarial patch is generated by increasing the efficiency loss defined by different efficient ViT. The experiments on ImageNet and CIFAR-10 demonstrate the effectiveness of the proposed attacks.

**Strengths:**

1. Studied an interesting and important problem: energy efficiency of ViT inference
2. Demonstrated that universal patch attacks against energy-efficient ViT are effective.
3. Nice figures to demonstrate why the attack works.

**Weaknesses:**

1. Limited technical novelty. The attack algorithm is a straightforward combination of several well-known techniques. (1) To generate an adversarial image, the attacker defines an attack loss and uses gradient-based optimization. The attack loss is trivial to define, since each efficient ViT algorithm already proposed a loss for computation efficiency. (2) To achieve universality, the attacker performs the optimization across different images – the most common strategy for universal attacks. (3) The adversarial patch threat model has also been studied extensively before. (4) the idea of using adversarial examples to attack computation efficiency is not new either. The only major contribution I see from this paper is to experimentally demonstrate that energy attacks against ViT do exist.
2. The paper assumes that FLOPs determine energy consumption but does not provide evidence or reference (though it is likely that this correlation between FLOP and energy consumption exists). Also, the paper does not provide details about how FLOPs are estimated in evaluation.
3. The paper argues that the patch attack is a more realistic attack because we cannot modify every pixel for systems like a robot. However, the patch generated in this paper is still in the digital space and is unlikely to work in the physical world. This questions the necessity of using the patch attack threat model.

**Questions:**

1. what are the new challenges encountered by SlowFormer and what are your corresponding solutions?
2. why is it necessary to use the patch threat model while all experiments are still in the digital domain (not physical-world attacks)?

---

### Official Review · Reviewer_fHNd · 2023-10-31

**Soundness:** 3 good
**Presentation:** 2 fair
**Contribution:** 1 poor
**Rating:** 3
**Confidence:** 4

**Summary:**

The paper explores the vulnerability of adaptive efficient methods to universal adversarial patch attacks. The experiments show that even a small patch can significantly increase the compute and power consumption of deep models. Moreover, they also propose a defense method that can reduce the success of such attacks.

**Strengths:**

- The paper provides some new results of the vulnerability of efficient Vision Transformers,

**Weaknesses:**

- The idea is straightforward, and the novelty is limited.
- Previous works have shown that efficient vision transformer methods are vulnerable to adversarial attacks. What is the significance of developing an adaptive patch to a universal patch? What is the major challenge?
- What is the major contribution of the proposed method?
- Regarding defense, I believe standard adversarial training should not be seen as a contribution.
- Regarding assumption, the paper notes that they did not experiment with real hardware to measure power consumption.

**Questions:**

Instead, they assume that the power consumption is proportional to the number of FLOPs and report the change in FLOPs of the inference time. Is this assumption proved by previous works?

---

### Official Review · Reviewer_CMkn · 2023-10-31

**Soundness:** 3 good
**Presentation:** 2 fair
**Contribution:** 2 fair
**Rating:** 5
**Confidence:** 4

**Summary:**

This paper proposes an adversarial patch attack that offsets the computational savings provided by efficient vision transformers. Those models work by reducing either the number of tokens they process or patches, self-attentions, and transformers they use during forwarding. The attack generates adversarial patches by solving optimization problems over the training samples, and the objective is the negation of the losses used for the dynamic inferences. In evaluation with the three adaptive ViTs, the attack offsets computational savings (100% on A-ViTs and 40-60% on ATS and AdaViT). The attack success against A-ViT has not been affected by the patch size and location. The paper proposes an adaptation of adversarial training to mitigate this threat.

**Strengths:**

1. The paper proposes an adversarial patch attack against energy-efficient ViTs.
2. The paper demonstrates its effectiveness against three adaptive ViTs.
3. The paper adapts adversarial training and proposes a countermeasure.

**Weaknesses:**

1. The proposed attack seems a bit incremental, given a body of prior work on this topic.
2. The attack pursues "universality" by exploiting a patch, but still, different ViTs should be attacked with different methods (i.e., different crafting objectives).
3. The ablation was performed on the most successful case, possibly misleading readers.
4. The adversarial training may empirically work, but it loses "provable guarantees."


Detailed comments:


I like the research direction where the paper wants to examine the computational robustness of adaptive ViTs. It is also interesting to observe that such adaptive methods are not robust to an adversarial patch that only takes up 8% of input images. However, even if the direction is interesting, I think the paper is half-baked. Here are my evaluations of the paper's weaknesses, and I hope they will be addressed in the next submission (ICML 2024).


[Incremental; Novelty Over the Prior Work]

Given there have been many works demonstrating the weaknesses of adaptive deep neural network computations, my impression is that this work is incremental. Compared to the prior work, the differences I see are: (1) Not the adversarial attack, but the adversarial patch attack. (2) Not attacking convolutional networks, but transformers. From the current paper, I don't see any particular scientific reasons why these differences are important, and I believe that the papers with "accept" quality must have those points addressed.


[Partial Universality of Patch Attacks]

The paper follows the philosophy behind universal adversarial attacks: a patch that can negate the computational savings of adaptive vision transformers. However, the attack (proposed) still needs to be computed for each mechanism. In the prior work on universal attacks, they generally demonstrated a (universal) patch that can fool any downstream models.

(If this is not possible in the future experiments) I believe this observation is also particularly interesting as the three adaptive mechanisms this paper examines rely on different things (which I don't know what, but I do believe that finding out such differences and their attributions to the weaknesses is essential).


[Weak Ablations]

The current ablations are performed on the most successful attacks. But as the attacks are less successful against ATS and AdaViT, I am more interested in the sensitivity of the proposed attacks on the two models. Showing only on A-ViT can mislead readers (even if there is no such intention).


[No Guarantee, Adversarial Training]

Adversarial training is a "provable" method to provide robustness against adversarial input perturbations (e.g., when it is bounded within the l-2 norm). However, the proposed adaptation, even if the adaptation may reduce the attack's effectiveness, the new adversarial training does not provide any "provable" guarantee. This is particularly a problem because there can be a single case where the new robust models are not resilient to this proposed attack.

I also believe that encouraging the community to develop defenses while this paper only talks about adaptive adversarial training is not an ideal way to conclude the paper. The paper (I believe) should provide a discussion about several defensive approaches (their downsides and strengths) against the proposed patch attack so that the community can start future work.


[Presentation]

1. The paper talks about unrelated things (or some speculations) in Intro, such as battery life.
2. In Page 5, the paper mentions "train the patch," does it mean crafting an adversarial patch?


Overall, for those reasons, I am leaning toward rejection.

**Questions:**

My concerns and questions are in the weakness section.

**Details Of Ethics Concerns:**

No ethics concern

---

### Official Review · Reviewer_Mjw1 · 2023-11-04

**Soundness:** 3 good
**Presentation:** 2 fair
**Contribution:** 3 good
**Rating:** 5
**Confidence:** 4

**Summary:**

This paper explores an adversarial patch that increases model energy consumption. The authors found that some models based on dynamic calculations can adaptively adjust the model's computational load based on inputs. Therefore, the authors believe that an adversarial patch can be used to maintain the maximum calculation amount of the model, which can increase the overall power consumption. This is called an energy attack. In this paper, the authors proposed a universal patch attack against three dynamic ViTs and experimentally demonstrated that it is effective.

**Strengths:**

1. The authors explore the adversarial robustness on dynamic ViT.

2. It is an interesting idea to improve the power consumption of the model through adversarial patches.

**Weaknesses:**

1. The proposed attack method is designed for a specific dynamic ViT structure and lacks generalization. Authors may consider proposing a paradigm to generalize this.

2. The generated adversarial patches are not tested in front of different models and cannot reflect generalization.

3. Our biggest concern is that the AI model does not account for a high proportion of power consumption in the drone scenario described by the authors. Dynamic ViT also doesn't appear to be widely deployed on devices.

**Questions:**

See Weaknesses.